# "It's just an issue and you deal with it… you just deal with it, you move on and you do it together.": Men's experiences of bacterial vaginosis and the acceptability of male partner treatment

Rebecca Wigan[1]*, Cathy Vaughn[2], Lenka Vodstrcil[1,3], Michelle Doyle[1], Marti Kaiser[1], Collette McGuiness[1], Catriona S. Bradshaw[1,3‡], Jade E. Bilardi[1,3,4‡]

1 Melbourne Sexual Health Centre, Alfred Health, Melbourne, Victoria, Australia, 2 Melbourne School of Population and Global Health, The University of Melbourne, Melbourne, Victoria, Australia, 3 Central Clinical School, Monash University, Melbourne, Victoria, Australia, 4 Department of General Practice, The University of Melbourne, Melbourne, Victoria, Australia

‡ These authors are joint senior authors on this work.
* rwigan@mshc.org.au

**Data Availability Statement:** Due to the small sample size and the interview transcripts

## Abstract

Bacterial vaginosis (BV) is a common vaginal infection among women of reproductive age. Increasing evidence suggests BV may be sexually transmitted indicating a potential role for the treatment of sexual partners. If partner treatment reduces BV recurrence in women, real-world success will depend on sexual partners' willingness to accept it. However, a lack of data exists on the acceptability of partner treatment among sexual partners, and no data exists on male partners' experience of BV specifically. The aim of this study was to explore male partners' views and experience of BV and their attitudes toward associated partner treatment. A social constructionist approach informed the framework of this study. Semi structured interviews were conducted with eleven men who participated in a BV partner treatment trial. Interviews were transcribed verbatim and analysed thematically. In the absence of symptoms in themselves, BV had little impact on men beyond their concerns for their partner's health and self-esteem. Acceptance of treatment was largely a demonstration of care and support. While all participants had accepted treatment, men surmised the primary reasons other men may reject treatment as being: if they felt BV had "nothing to do with them", which was related to not wanting to be viewed as having a 'problem' and exacerbated by norms of masculinity and STI-related stigma; lack of a diagnostic test to indicate if a male "had BV"; and a casual or less established relationship. Men's attitudes to BV and partner treatment were primarily influenced by the nature of their relationships. The ambiguous aetiology of BV appears to attenuate STI related stigma and questions of infidelity.

containing sensitive and potentially identifying information, the ethics committee has not approved public release of this type of data. Interested researchers may contact Kordula Dunscombe of the Alfred Hospital Ethics Committee if they would like access to the data (k. dunscombe@alfred.org.au).

**Funding:** The author(s) received no specific funding for this work.

**Competing interests:** The authors have declared that no competing interests exist.

## 1. Introduction

Bacterial vaginosis (BV) is the most common vaginal infection in reproductive aged women globally [1]. It has been estimated that anywhere between 5–70% of women may be affected by BV at any given time [2–4]. Unfortunately, the cause of BV is unclear as it is not the product of a single pathogen, but it involves a shift in the composition of the vaginal microbiota to an unhealthy state dominated by BV associated bacteria (BVABs) [5]. The underlying reason for this change is not yet fully understood [6]. Symptoms of BV occur in more than 50% of women and most commonly include a thin homogenous vaginal discharge and malodour often likened to a "fishy odour" [7]. Left untreated, BV is associated with serious adverse sequelae including; an increased risk of transmitting and acquiring HIV [8], pelvic inflammatory disease, miscarriage, pre-term delivery and low birthweight [9]. Current first line treatment for BV is a seven day course of oral or vaginal antibiotics, however recurrence rates are high with more than half of women experiencing recurrence within six months [10,11].

While BV can result in adverse physical sequelae, several studies have also shown the symptoms of BV often have a significant impact on women's psychosocial well-being and relationships [12–14]. Overwhelmingly, the most distressing symptom for women is the malodour, which makes them feel "dirty", "unattractive" and "ashamed" [12–14] and consequently negatively impacts on their self-esteem and sex lives [12–14]. This largely centres around women's fear that others, and in particular their sexual partners, will notice "the smell" which results in a reluctance to engage in sexual activity and often undermines their feelings of intimacy and closeness with their partners, even if partners are supportive [12,13]. For some women, the shame felt around BV symptoms is attributed to broader societal stigma attached to women's sexuality and STIs [13,14].

Given its high global prevalence, associated morbidity, high rates of recurrence, and adverse psychosocial impacts, identifying new approaches to improve sustained cure for BV in women is of significant public health importance.

The concept that BV may be sexually transmitted, with incident and recurrent infections due to exposure to infected sexual partners, has been a controversial area of inquiry for the last 50 years. Currently, BV is not considered an STI and treatment of partners is not routine practice in any clinical guideline. While there is strong evidence supporting a dominant role for sexual transmission in the pathogenesis of BV, other factors may also contribute, although BV is rare/absent in women prior to sexual debut [15]. These factors include racial variants in vaginal microbiota, altered host immunity, an endogenous source, environmental factors such as nutrition and intravaginal practices such as douching [16].

Determining whether BV is sexually transmitted has been complicated by difficulties in determining its aetiology, lack of clear evidence of infection in males, and the failure of five out of six partner treatment trials to demonstrate an impact on recurrence [17–24]. However, a systematic review of all partner treatment trials to date found them to be inconclusive by current standards, due to the use of non-standard BV treatments, no measure of adherence, poor retention and small sample sizes; and recommended larger trials of first-line therapies be conducted [18]. Furthermore, recent advances in epidemiological and molecular evidence strongly support the potential for sexual transmission of BV. Studies have found that women who have an ongoing male sexual partner are twice as likely to experience BV recurrence [3,10] and that inconsistent condom use for penile-vaginal sex is also associated with higher rates of recurrence after treatment [10]. Male carriage of an integral BV Associated Bacteria (BVAB) (*Gardnerella vaginalis*) is commonly reported [25] and a recent study has demonstrated BVAB colonization of the penis to be significantly influenced by sexual activity and circumcision

[26]. These advances in knowledge provide strong support for the sexual transmission of BV and potential role for male partner treatment (MPT) to improve sustained cure in women.

Currently, researchers at the largest free sexual health clinic in Victoria, Australia, the Melbourne Sexual Health Centre (MSHC), are conducting the StepUp Randomised Control Trial (StepUpTrial; Australian and New Zealand Clinical trial registry: ACTRN12619000196145). This is a well-powered MPT trial of treatment of women and concurrent treatment of their male partners with combined oral and topical antimicrobial therapy. The RCT aims to determine whether MPT improves BV cure and reduces associated sequelae in women. If successful, this will represent a major shift in the way BV is managed and, for the first time, provide an opportunity for the long-term control of this infection that has such significant health, economic and personal costs [27]. However, if MPT is shown to be effective, practical success within heterosexual couples will depend almost entirely on the willingness of men to accept and adhere to treatment. In past MPT trials, adherence of men to partner treatment appears to have been assumed, as reporting of this variable has been largely absent.

Though considerable qualitative literature regarding attitudes to partner *notification* for STIs has emerged over the last 20 years, there is a dearth of literature exploring reasons for partner *acceptance or decline* of STI or BV treatment. Existing evidence suggests that men are more likely to delay or avoid seeking treatment, even for a diagnosed STI, if they perceive the infection to be of little personal health consequence [28,29]. As the presence of BVAB on the penis is not believed to commonly cause BV related symptoms or sequelae in men, it is possible such attitudes may undermine the success of MPT for BV.

To date, there is a lack of data on men's views and experiences of BV within their relationship, and their attitudes to the possibility of being offered partner treatment. The aim of this study was to explore male partners' experiences of BV and the acceptability of associated MPT for BV. If MPT is found to be effective, a nuanced understanding of what motivates men to accept or decline treatment will be crucial to the successful uptake of MPT. This study adds to the extremely limited qualitative literature of partner treatment acceptance for genital infections more generally. This will be the first study to provide an insight into how BV infections affect men, developing a body of knowledge that provides a holistic view of BV's impact on both partners in a sexual relationship.

## 2. Methods

This study has been reported in accordance to the Consolidated criteria for reporting qualitative research (COREQ) guidelines [30]

### 2.1. Ethics statement

Ethical approval for this study was granted by the Alfred Hospital Ethics Committee, Victoria, Australia, Application Number 506/18 on the 27th November 2018.

### 2.2. Theoretical framework

A social constructionist methodology informed this study, as partner treatment is a function of a social relationship by its very nature. Social constructionism fits within the research paradigm of interpretivism which seeks to understand 'the nuanced world of lived experience from the point of view of those who live it' [31]. From the social constructionists' perspective, individuals' perceptions of reality and the meaning of phenomena are informed by the social and cultural norms operating at the time and within that context [32] . Such perceptions of reality can be greatly influenced by the meaning society attributes to an illness or condition [33], as well as social constructs of gender [34,35].

Despite not being classified as an STI, the genital location of BV means it is likely to be similarly understood. STIs are highly stigmatised conditions, and the negative social connotations of genital infections are likely to influence how people perceive, cope with and respond to a diagnosis and treatment of BV.

Theories of gender and health argue that health-related beliefs and behaviours are a means for demonstrating masculinity and are therefore culturally understood symbols that can be leveraged to assert a particular identity or cultural norm [35–37] Traditional masculine ideals that prescribe stoicism, dominance, independence and self-reliance [35–38] are increasingly identified as one of the most important socio-cultural factors influencing men's health attitudes, behaviours and outcomes.

### 2.3. Method, research team and reflexivity

Semi-structured interviews were utilised to explore lived experiences from the first person point of view [39]. This permitted the researcher to target key areas of research interest, whilst allowing the flexibility for unexpected themes to emerge from men's lived experience.

All interviews were conducted by RW, an experienced sexual health and research nurse at MSHC who undertook the research a part of her Masters of Public Health. As a sexual health nurse, RW is highly accustomed to speaking to people about sexual health and has a sound understanding of the epidemiology of BV. However, it is acknowledged that RW's gender (cisgender woman) and profession may have influenced the nature of information participants shared. RW had no prior relationship with participants and is not part of the StepUp studies' recruitment team. Men were informed that the research was being undertaken to better understand men's perceptions and experiences of BV, as well as motivations for the acceptance or rejection of male partner treatment.

### 2.4. Recruitment and participant selection

This study was conducted at Melbourne Sexual Health Centre, Victoria Australia.

Men were purposively recruited through the StepUp pilot and RCT (StepUp studies) as it allowed access to men who had been offered partner treatment for BV, something which is not yet standard clinical practice.

Within the Step Up studies, male partners of women being treated for BV were randomised to oral Metronidazole and a topical antibiotic cream applied to the penis twice a day for 7 days or current standard of care (female treatment only). Participants self-collected genital samples monthly for a follow up period of 3 months. All male procedures, including initial recruitment were able to be conducted remotely via phone and using post packs. To ensure consistency, all male partners were provided with the same information regarding the trial and BV at recruitment; that BV is a commonly occurring infection in women, that BV associated bacteria have been found on male genitals and that these bacteria may be exchanged during sex. No further education was provided. Men were also made aware that all answers in questionnaires were kept strictly confidential and not shared with partners.

As men must have been offered the opportunity to participate in the StepUp studies to be eligible for interview, with the eligibility criteria for this outlined in Table 1.

All men who were still actively engaged in the StepUp pilot at the time of ethics approval were contacted via a single SMS offering the opportunity to participate in the qualitative study. Men were invited to contact the study number if they were interested in participating. If they did not respond to the SMS, no further contact was made.

Men who participated in the RCT could indicate their interest in being interviewed by checking a tickbox on the StepUp RCT consent form. If men checked this box, their contact

**Table 1. Study eligibility and sampling framework.**

| Eligibility of male participants for Step Up Trial |
| --- |
| • Female partner with a confirmed clinical diagnosis of BV who has been offered participation in either the Step Up Pilot or Step Up RCT |
| • Aged 18 years or above |
| • Monogamous relationship with female partner for minimum of 8 weeks |
| • Sufficient English proficiency to understand study requirements |
| • Able to comply with study requirements |
| • No contraindications to treatment |
| **Additional eligibility for qualitative Male Partner Treatment study** |
| • Had been offered participation in Step Up studies |
| • Consented to audio recording of interview |

As described, this study aimed to recruit men who had either accepted or declined MPT through the StepUp studies. Men were recruited in one of three ways.

details were forwarded to RW. All men who provided consent were contacted by RW until saturation was deemed to be met.

Men who declined participation in the StepUp studies were contacted through their female partners who were provided with a flyer including a summary of this study and the research telephone number. Men were invited to contact the study telephone number if they were interested in participating.

RW contacted all men who had indicated their interest to explain the study in detail and arrange an appropriate time for interview. Men were contacted a maximum of three times before being considered lost to follow up.

## 2.5. Data collection

Men were offered the option of a face-to-face interview at MSHC or a telephone interview. Participants were provided with a Participant Information and Consent Form (PICF) prior to the interview which assured them participation was voluntary and that any data they provided would be treated confidentially. Men were informed their data would be stored on double password protected on-site databases, and no identifying information would be used in any publications arising from the study.

Men were informed that the interviews needed to be audio recorded to allow the interviewer to pay full attention to their accounts and lived experience of being offered BV treatment. No men raised concerns about the audio recording or declined to participate in the interview based on this requirement. Men were informed they did not need to answer any questions they were not comfortable with and could cease the interview at any point.

All participants chose to participate in a phone interview. Verbal consent was provided whereby RW read the previously provided consent form aloud and asked them to state their name, the date and affirmation of their consent. This was recorded on a separate audio file to their interview data to ensure confidentiality. No names were used during the interview.

All interviews were conducted between January and September 2019. Prior to interview, men were sent a copy of the PICF to ensure they had a full understanding of the study.

At the commencement of the interview, men were asked a number of demographic questions including length of their current relationship, prior STI testing and couple contraceptive use. Men were then asked a series of open-ended questions exploring men's knowledge of BV, views and attitudes about STIs, the impact of BV on their relationship, motives for MPT

acceptance and hypotheses why men may decline MPT. All interviews were audio recorded. Men were not reimbursed individually, but went into a ballot to receive a $200 gift card for their participation.

## 2.6. Data analysis

RW met regularly with JB and CV, experienced social researchers adept in qualitative methods in the area of sexual and reproductive health, to discuss the interviews and emerging themes. New lines of questioning were added throughout the data collection period including questions around the influence of uncertainty of infection in men on their willingness to accept MPT, and experience of personal symptoms (refer to Table 2). When RW felt data saturation was approaching, she again met with JB to review the interview transcripts and discuss emerging results. At this time, after eleven interviews were complete, it was decided that data saturation had been met and no further interviews were required.

All interviews were transcribed and de-identified to protect confidentiality. An inductive latent thematic analysis [40] was undertaken. Analysis commenced with immersion in the data through transcribing all interviews, and reading and re-reading of transcripts. RW took notes with ideas for potential codes and themes throughout. Once familiar with all aspects of the entire dataset, RW manually generated initial codes inductively by identifying and highlighting key features in the dataset. Transcripts were then imported to N-Vivo 12 [41] for data management. Codes were sorted into potential broader themes and sub-themes and data examined for similarities and differences. The process of coding and formation of themes was a circular, ongoing process to ensure they provided an accurate representation of the data set. JB and CV reviewed a subset of transcripts to confirm coding and thematic analysis. No differences in interpretation were evident and the final interpretations of the data were confirmed.

## 3. Results

During the recruitment period, before it was decided data saturation had been met, 20 men were eligible for this study. All eligible men were contacted for interview. Of those contacted, 11 men were interviewed; 7 were recruited from the StepUpPilot and 4 from the StepUpRCT. Of the remaining 9 men that were contacted, 8 did not respond to the SMS invitation and 1 decided to withdraw after consenting. These 9 men were not followed up further as per the study protocol and ethics approval. If data saturation had not been met after 11 interviews,

**Table 2. Interview schedule.**

| Interview Schedule Topics |
| --- |
| • Knowledge of BV |
| • Prior experience of BV |
| • Impact of BV on self, partner and relationship |
| • BV as an STI |
| • STI experience and attitudes |
| • Reasons for participation in the trial |
| • Experience of study and treatment |
| • Reasons they or other men would decline treatment and/or the trial |
| • How would they like to receive information about BV |
| **Revised during data collection to include further questions about:** |
| • Impact of uncertainty of "having the infection" |
| • Symptoms in men |

further participants would have been prospectively recruited from the men enrolled in the ongoing StepUp RCT. All participants had received treatment, although two were yet to take the antibiotics at the time of the interview. No men who had declined participation in the StepUp studies contacted the research team to participate. As men could only be invited through their female partners who were not obliged to provide them with study information, it is not possible to know how many men who declined partner treatment were offered the opportunity to participate.

All participants chose the option of a telephone interview, which lasted between 20 to 60 minutes (average 39 minutes). Participants ranged in age from 23 to 60. All participants were in committed relationships and had been with their current partner from 4 months to 8 years, with half (n = 5) in relationships under a year in length. All had previous STI testing experience. Further participant characteristics are outlined in Table 3.

Table 4 presents three case scenarios of participants and provides an overview of the differing experiences of men in this study, both in terms of the impact of BV and experiences and reasons for accepting and declining partner treatment.

## 3.1. How men understood BV

Overall, men had little awareness of BV but had developed an understanding of the infection due to their participation in the StepUp studies.

## 3.2. Men's lack of BV awareness

Generally, men had low levels of awareness and knowledge about BV, unless they had previously been with a partner who had experienced the infection. Most men reported that prior to their partner's diagnosis, they had never heard of BV. Men whose partners had previous

**Table 3. Participant characteristics (N = 11).**

|  | N or Median [Range] |
| --- | --- |
| **Age** | 28 [23–60] |
| **Identified as heterosexual** | 10^ |
| **Born in Australia** | 10 |
| **Education level** |  |
| Secondary school to year 12 | 6 |
| TAFE diploma or certificate | 1 |
| Undergraduate degree | 2 |
| Postgraduate degree | 2 |
| **Couple contraceptive methods**\* |  |
| Condoms | 7 |
| Implanon | 3 |
| IUD | 2 |
| Oral contraceptive pill | 1 |
| Vasectomy | 1 |
| **Length of relationship** | 18 months [4–96 months] |
| **Prior STI testing experience** | 11 |

^ the remaining participant identified as bicurious, which is a heterosexual person who is interested in having a sexual experience with a person of the same sex [42]

\*May total more than 11 as more than one method of contraception use

**Table 4. Case scenarios.**

**The impact of male symptoms**

Participant 10 was a man in his thirties who had himself experienced a "fishy" genital odour for over ten years. Symptoms would increase with any sexual activity and his sexual partners had also experienced many repeat BV infections, including his previous wife and current partner. This led him to believe his symptoms could be sexually transmitted, but this idea was previously rebuffed by his female partner due to BV being a "women's issue". The odour he experienced had a substantial effect on his sexual confidence, undermining his willingness to embark on new sexual relationships or receive oral sex due to fear of "the smell" being discovered. Despite the impact of his symptoms on his personal life, he came to understand "the smell" as "normal" and "just the way he was". It was his care for his partner and witnessing her distress with BV that led him to investigate if there was any new research on sexual transmission of BV from men, as this continued to be his assumption. Had his partner not demonstrated concern about her own symptoms, he would not have sought further medical attention for himself. For this participant, who experienced a complete clearance of his symptoms following MPT, this prospect of 'cure' for his own symptoms was revolutionary. *"It's massive!. . .that's something that I've been struggling with for probably more than 10 years. . .hopefully now, that'll (clearance of symptoms) help to break down some of those internal barriers to that (receiving oral sex)."*

**From declining MPT to accepting MPT**

Participant 3 was a man in his twenties who had been in a relationship with his partner for over five years. His partner had asked him to participate in a similar MPT trial for BV at the beginning of their relationship, but he had declined at that time. His reasons for previously declining were: 1) He did not believe that BV had anything to do with him as he had no symptoms and had a clear previous STI screen—"it didn't apply to me" 2) The relationship was relatively new—"we're not that far into our relationship, so why do I need to do this for you?" 3) He simply did not want to–"I didn't wanna do it, so I didn't do it". These reasons reflect the hypotheses of all participants around why men may not accept treatment. On this current occasion of being offered MPT, the more established nature of his relationship, and possibly the recurrence of his partner's symptoms, were sufficient to lead to treatment acceptance despite no more evidence that his partner's BV "had anything to do with him". This demonstrates the central importance of the nature of a relationship (and how committed it is) in influencing MPT acceptance. Despite accepting treatment, this participant demonstrated an extremely judgmental attitude towards STIs and those who had experienced them, as well as demonstrating high levels of the type of "manly" masculinity described by all participants as a likely barrier to MPT acceptance. For this participant, the concept of BV as a "lady problem" was protective against the "crushing" social anxiety that he felt treatment for an STI would result in.

**Pragmatic and caring**

Participant 1 was a man in his twenties who had been in his relationship for six months. He had noticed symptoms in his partner of slight odour and her discomfort during sex. The communication in the relationship was very open, leading to the couple seeking medical attention together to identify the cause of his partner's symptoms. He was very receptive to the possibility that he could be a "carrier" and keen to have treatment to help and support his partner as he could "see the discomfort she was in". He saw this as his role as a partner and believed that the support he had demonstrated for his partner through the process had strengthened their relationship. His partner had also compared the higher degree of comfort she experienced "going through this with him" than she would have with previous partners who would have been less supportive. This demonstrates the importance of female comfort in engaging men in MPT, as well as the role of the relationship and care.

Table 4. Case scenarios demonstrating different impacts and experiences of BV and partner treatment.

experience of BV were more likely to show some knowledge of BV and less inclined to feel confused or frustrated as they knew it was a treatable condition.

> *". . .it wasn't a freak out coz she kind of knew what it was . . .but maybe for someone who doesn't know what it is. . . there'd be a lot of confusion. . . .I guess for a male, he kind of doesn't know what's going on. . .if she doesn't know what's going on, how's he going to know what's going on?"—Participant 4 (24 years)*

## 3.3. BV as an STI

Men in this study were informed as part of the BV studies that BV was not currently classed as an STI, however, increasing evidence suggests it is associated with sexual activity and that bacteria may be "exchanged between partners".

Participants' views around STIs varied. Some men viewed them as a part of life that one has to be proactive about identifying and addressing, whilst others saw an STI diagnosis as a personal failing that would "crush" them and endanger their standing in social networks.

*"I'd be crushed. I'd be like. . . 'Oh you obviously haven't been listening to the sex ed teacher very well, have you mate.' So, I'd be pretty upset if I got one of them."–Participant 3 (28 years)*

Interestingly, as BV is not currently classified as an STI, men who demonstrated high levels of STI stigma did not express the same degree of distress, stigma, or fears of infidelity around BV that an STI diagnosis would evoke.

*"I don't think BV would have something so bad (as an STI)."- Participant 3 (28 years)*

A number of men mentioned that "other men" may suspect their partner had been unfaithful due to the sudden appearance of symptoms, yet they did not. However one participant reported that if BV were classified as an STI, his partner's diagnosis would have made him question her fidelity. This demonstrates that the impact of BV on men within their relationships centres to a great degree on how they understand the infection.

*"If she came to me and said that she had herpes or something like that I'd probably be asking a few questions because it's. . ... you know. . ., from what I know. . . like it's not an STI. Like it's, you know what I mean? I don't think she's been cheating on me. . . But you know, when your missus comes up to you and is like, 'Oh, I got tested and I've got vaginosis'. It's like, you know, honestly, I'd feel the same way if she came to me and said she had tonsillitis."—Participant 5 (34 years)*

Whilst men generally spoke of not wanting their social networks to be aware of being treated for BV as this may be "embarrassing" and leave them open to ridicule, this was somewhat minimised by the current classification of BV. This suggests that the current conceptualisation of BV may be protective, despite a number of men identifying that STI stigma could be applied regardless of the classification of BV purely because of its genital location. Indeed, one participant identified that the "flexibility of the knowledge" could protect individuals from the stigma of STIs.

*"That classification (STI) can obviously scare some people. But when . . . we're not too sure what category to put it into, it's a bit more open I suppose. There's a bit more flexibility with the knowledge. . .but I suppose once it does go into that category of an STI, some people might look at it in a different way. . .I think people's views would change."–Participant 6 (26 years)*

### 3.4. The impact of BV on men

Overall, most men reported that BV did not directly impact on them physically or emotionally, beyond concern for their partner's health and psychological wellbeing and the impact it had on their sexual relationship due to their partner's concerns around symptoms.

### 3.5. Physical impact

Most men in this study felt BV had no direct impact on them physically as they did not experience any symptoms.

*"Well I don't think it has affected me at all. I've never been aware of any symptoms."–Participant 2 (60 years)*

Interestingly, however, two men spoke of experiencing genital symptoms that they attributed to BV. One participant spoke of developing symptoms of genital discomfort and sensitivity after occasions of unprotected sex with his partner who had long standing symptoms of BV that she had been unable to have accurately diagnosed and managed.

*". . . she'd had this for quite a long time and she'd had it somewhat diagnosed by a few different GPs. . . one had said that it was BV while another would say that it's not. Now we've gone backwards and forwards, and I encouraged her to follow it up further. . . It seemed to be more something she'd been experiencing for a long time, and I just happen to come onto the scene and then be affected by it. . ."–Participant 11 (23 years)*

Another spoke of having experienced a genital "fishy like" odour for *"as long as (he) could remember"* and had assumed he may be the source of his partner's infection when she developed similar symptoms (see Table 4: Case scenario 1).

*"I always suspected that I was carrying BV and so when my partner complained about the smell like a fishy smell. I sort of thought, cuz I had experienced that on myself and my partners in the past. And always just have the assumption that people could carry. And she said, 'No, no, it can't be you. It has to be me.' And so I was kind of confused by that."–Participant 10 (35 years)*

## 3.6. Psychosocial impact

Men who felt they had experienced symptoms, were more likely to report personal impacts centring around blame, reduced self-esteem, and negative impacts on their sexual relationships and sex lives.

For the participant who had experienced a "fishy" odour for ten years, the impact of this symptom had a profound *personal* impact on his sexual confidence. His embarrassment and fear of sexual partners becoming aware of the odour prompted long lasting changes in hygiene practices and sexual behaviour (see Table 4: Case scenario 1).

*"It's the smell. Just the smell, really. . . I'd have to make sure that I was like, really clean and the smell wasn't there. Coz obviously it would be embarrassing otherwise. . . It was always something that I was, I would go in concerned about. It affected my desire to have oral sex, or be able to receive oral sex- definitely! I think it's probably been the cause of me having almost a dislike of oral sex."—Participant 10 (35 years).*

The other participant's symptoms did not undermine his own self-esteem but instead affected his attitude towards his partner and the relationship due to the belief that his partner had "given him something". This generated unwanted and uncomfortable feelings of *blame* towards his partner and anxiety around sexual activity, which created further tension in the relationship. He noted that if he had not experienced symptoms, BV would not have played such a significant role in his relationship despite the presence of his partner's symptoms.

*"It definitely led to some strain. . . if I were more resistant to the symptoms, this may not have been something that would be so significant in my relationship. . .or um, life with my partner."–Participant 11 (23 years)*

**3.6.1. The impact of female symptoms.** Half of the men interviewed had not noticed any symptoms in their partner, while others most commonly noticed the malodour with descriptions ranging from "not a big deal" to "unpleasant". Most men reported that their partners seemed more aware of the odour than they were and their partner's embarrassment tended to have a negative impact on their sex lives and at times, caused tension and frustrations within the relationship.

*"Yeah, I guess my partner would become withdrawn sexually. And if I would like to engage in sexual activity, she would be too embarrassed, and I guess that would sometimes cause a rift in the relationship."–Participant 10 (35 years)*

**3.6.2. The impact of sexual withdrawal and partners' diminished self-esteem.** Men did not generally see BV as a "big deal" in and of itself, but the impact of it on women's self-esteem and confidence was of great concern. Many men identified that the impact of symptoms on their partners was significant, and led women to feel unattractive and sexually inadequate. This in turn affected women's self-esteem, and men often felt a sense of helplessness around wanting to help their partners feel better, but not knowing how. Indeed, it was clear that the impact of BV on men centred largely around their partner's response to their symptoms rather than the presence of symptoms themselves.

*". . .the main thing would be her feeling a lot more self-conscious about it. And I think not even, I think even outside of sexuality, in her confidence generally which did flow on to sexuality as well."–Participant 8 (31 years)*

**3.6.3 Positive impact of BV and partner treatment on the relationship.** While men described various negative impacts of BV, many identified that the appearance of symptoms led them as a couple to improve their communication about sex and sexual health. The greatest positive impact however, was unquestionably the "bonding experience" of partner treatment which almost all men believed further strengthened their relationship.

*"If anything, it's (BV treatment) probably just binded us closer a little bit more. And just something else that we've been able to tackle . . . if anything, just stronger as a couple."–Participant 6 (26 years)*

## 3.7. Lived reasons for MPT acceptance

All participants had accepted MPT and the main reasons they had accepted treatment centred on feeling it was part of being in a committed relationship, their role as a supportive partner and man, part of open communication in a relationship and concern around potentially being the source of infection.

### 3.8. Committed relationships–treatment as support

The most important reason men accepted MPT was unquestionably to show care and support for their partner. Men largely viewed treatment as an avenue to support their partner and as a way to demonstrate their commitment to the relationship. Whilst some men related the desire to "help" their partner with the distress her symptoms seemed to cause her, the act of taking treatment was a symbolic affirmation of their role as a supportive partner, with or without referring to this distress. Indeed, one participant described the acceptance of MPT as a metaphor for the relationship, where trivialising of BV would be a trivialisation of his relationship.

*"For how it affected my partner, well if I'm just gonna muck around with it, take it as a joke,.. I'm just taking the relationship as a joke at the same time."–Participant 4 (24 years)*

For a number of men, having the option of partner treatment was reassuring as it gave them something tangible they could do to address the issue of BV.

*"It would more be frustrating if there wasn't anything I could do about it."–Participant 7 (26 years)*

Two participants were so motivated to help their partners who were frustrated with repeat BV infections, they proactively sought out further information and ways to help, such as the StepUp studies.

*"My new partner, she was trying everything, and it was really stressing her out. And so I wanted to try and help her. And I guess that that was what drew me to sort of look for, look for other avenues."–Participant 10 (35 years)*

### 3.9. My responsibility as a man

A number of men used the same language to say they felt it was their responsibility or role in the partnership as a man to support their partner. In this way, their sense of responsibility was linked to the nature of the relationship, but also male identity.

*"That's your role as a man, to support your missus. . .when the time comes, you man up. That means you're supportive. You support your partner."–Participant 5 (34 years)*

For a couple of men, this sense of male responsibility was more general and derived from a sense that women suffered disproportionately with reproductive health.

*"I feel like in general women get the short end of the stick with basically anything to do with sex and reproduction, so anything I can do as a man that is like vaguely helpful, I feel it's my responsibility to do."–Participant 7 (26 years)*

Interestingly, participants in this study often differentiated themselves from "other men", who they felt could be unwilling to support their partners, take responsibility for sexual health issues or aligned themselves with traditional homophobic ideals of masculinity.

*"Let's put it this way, if a gay bloke walked through the door (at a worksite), they'd quit. Me and my friends aren't like that. . .it's a new age of men. . ..apparently."–Participant 3 (28 years)*

### 3.10. Communication within the relationship–have to know to act

Open, trusting communication within the relationship was cited as an important element of treatment acceptance. However, a number of men noted that open communication could be hampered by women's fear of being judged for having BV.

*"If he doesn't know, he's never gonna know. If she's too scared to tell him, (they will) just continue to throw it around"–Participant 3 (28 years)*

Though some participants felt their comfort in communicating openly was dependant on the length of the relationship, others stated that this had been a key element of their relationship from the beginning. Open communication enabled discussion of sexual health matters, allowed men to understand the cause of their partner's distress, and often led to shared or supported decision-making about seeking medical help for BV.

*"Me and my partner are very open with each other. We don't really hold back from anything. So we quite happily shared the information with each other and we understand if it's causing one of us pain, we're there for each other and so the first thing we did straight away was to go and see her regular doctor."–Participant 6 (26 years)*

### 3.11. I do not want to be a source of infection

Most men who understood BV as something they could potentially be "giving" to their partners were very pragmatic in their acceptance of MPT as a way to "fix it".

*"Well if she did get it from me, well, let's get rid of it. Or do what we can to get rid of it. Both get treated."–Participant 2 (60 years)*

For some men, the belief that they could unknowingly be posing a risk to their partners meant that they viewed treatment as a moral obligation. Men who were concerned about being a source of infection generally expressed that they would likely accept MPT regardless of the nature of the relationship within which it was offered, as this was the "right thing" to do.

*"Like, it's kind of a shock but. . .that it (BV) could be that it is (sexually transmitted) and that if (my partner) had it and I didn't do a treatment of it either, then I'd be giving it back to her. . . I kind of find that would be kind of a selfish act."—Participant 4 (24 years)*

All bar one participant described the treatment as easy, and a reason why they would accept MPT again in the future.

*"it's such a minor, non-intrusive treatment that I don't think there'd be any circumstance that I'd be like 'No, screw you. You'll just have to deal with it and risk it'."–Participant 7 (26 years)*

### 3.12. Hypothetical reasons for MPT decline

Despite having accepted MPT as part of the StepUp studies, many participants expressed strong views on why "other" men may decline MPT and hypothesised on what may influence their own choice of whether to accept MPT if they were offered it again in the future. Of note, one participant had previously been asked by his ongoing female partner to participate in a

preliminary study by our group years prior, and had declined at that time (see Table 4: Case scenario 2). The main reasons men proposed why they or other men might refuse treatment centred around BV having nothing to do with them, if they were in a casual or less established relationship where they did not feel obliged, and not wanting others to know they were being treated for fear of stigma and judgement.

### 3.13. It's nothing to do with me!

The primary reason participants believed men would refuse MPT, was due to a belief that BV had "nothing to do with them". This was seen to be the product of a traditional masculine identity in which having a health "problem" may be perceived as "weak"; BV being understood as a "lady problem"; and the absence of physical symptoms or a confirmatory test for men.

**3.13.1. The "masculine" man–it's not my problem.**   Avoidance of treatment was cited by most participants as a way to prevent the spoiling of a "masculine" identity. They hypothesised that MPT acceptance could be viewed as an admission of "having something wrong with you". Thus, MPT rejection could be a way for men to avoid this perception and responsibility. Participants linked this to traditional norms of masculinity, where "having something" is seen as "weak" and a personal failure. In this way, men would be able to conserve their masculine identity and "pride" by distancing themselves from the infection.

> "If a male does accept it, they either think that they're getting looked at like they're the problem. Or they don't want to see themselves as that. . . it's just a pride thing. It comes up to not wanting to look weak."–Participant 4 (24 years)

Of interest, one participant identified that if BV is indeed something that can involve men, the name itself may be alienating and somewhat feminising, presenting another potential barrier for "masculine" men.

> "It's a shame that it's called bacterial vaginosis. I feel like if anything, attaching the word vaginosis to something that's potentially happening to a guy could potentially bring up something emotionally in some guys."–Participant 10 (35 years)

**3.13.2. BV is a women's issue—Where's the proof it affects us?.**   Many men observed that the perception of BV as a purely female condition would lead to men declining treatment simply because they did not feel that taking it would present any personal benefit. This is particularly true in the absence of any symptoms in themselves.

> "If someone was seeing a girl and obviously the condition was only affecting the female, then they might be hesitant to take the pills or use the cream, if it's not having any physical or visible effect for their body."–Participant 1(27 years)

The absence of symptoms in men and lack of a confirmatory test for males were generally viewed as important barriers to MPT acceptance. Despite having accepted treatment as part of a trial in an effort to address their partner's distress, some participants also felt that they would want more evidence of infection in themselves if they were to accept treatment in the future, particularly if they were in a casual relationship.

> "I think if it was like something they got tested for and it was like 'you have BV. It's non-symptomatic but if you take this treatment, then you won't pass it to your partners' then I think

*most guys would do that... but maybe if they had a more long term girlfriend as opposed to like a casual relationship or something."–Participant 7 (26 years)*

**3.14. Casual relationships—treatment as a favour.**   Participants viewed casual or less established relationships as an important reason that men may decline MPT. Whilst accepting treatment within a committed relationship is a demonstration of care, acceptance of MPT for BV in a causal relationship may be seen as a favour. This was reflected by one participant when explaining why he had declined a preliminary MPT study by our group years prior.

*"We're not that far into our relationship, so why do I need to do this for you?".–Participant 3 (28 years)*

The lack of "proof" of infection in themselves was viewed by all participants to be of greatest relevance in casual relationships.

*"If men don't show the symptoms, it's harder to ... you can't just say, 'hey this is what you've got'. I think especially in casual relationship, a lot of men won't want to own up to having that and therefore won't seek treatment."–Participant 8 (31 years)*

**3.15. It's not about the treatment–stigma.**   Men were instructed not to drink alcohol or have sex during the seven days of antibiotic treatment. Participants generally felt that the treatment and its related restrictions would not be a huge barrier to treatment acceptance, however as noted earlier, other people knowing they were being treated for BV could be.

*"It's got more to do with what it's (treatment) for than what you're actually doing."–Participant 8 (31 years)*

Participants believed that a fear of judgement from others would be an important reason men would decline MPT. Whilst being seen to "have a problem" could be perceived as "weak" in itself, the genital location of BV could evoke the stigma that surrounds STIs, leading to greater perceived judgement, ridicule, and shame. Thus, avoidance of MPT was a way to protect an inviolable, socially accepted identity.

*"People might take it as, like...'Ahh, you've got a disease' or something like that... like, they'd be like, 'There's something wrong with you. You've got an STD or something.' And, like, that would make it around the whole work site."–Participant 3 (28 years)*

## 4. Discussion

Research exploring the impact of genital infections on intimate relationships is extremely limited and has focused exclusively on the accounts provided by those with a diagnosed infection [43]. To date, three studies have specifically explored the impact of BV on women, all finding the infection to have a significant impact on women's intimate relationships and quality of life [12–14]. Whilst women provide accounts from their perspective of how BV affects their relationships and the responses of their partners, this is the first study we are aware of to speak directly to men about their views and experiences of BV and associated partner treatment.

This study has demonstrated that while the symptoms of BV often do have an impact on men's sex lives this appears to be mainly due to the shame and embarrassment women feel,

and men in this study demonstrated they are more concerned with the impact of symptoms on their partner's self-esteem than their sex lives per se.

It has also shown that many men are often unaware of their partner's symptoms and do not generally believe they are as noticeable as women believe they are. Women in other studies have reported receiving similar reassurances from their partners [13,44], yet this study is the first to demonstrate that this is actually a lived experience of men.

Men often noticed the distancing of their partners when symptoms were present, which reflects the finding by Bilardi *et al* (2013) that the experience of BV often undermines women's feelings of closeness to their partners [13]. The almost universal description of partner treatment as a "bonding experience" by the men interviewed demonstrates that the symbolic act of partner treatment (independent of treatment effect) has the power to bring couples closer, and improve relationship dynamics from a male perspective. This bonding aspect of partner treatment reflects some of the findings of another Melbourne-based study exploring experiences of partner notification for chlamydia in which some men felt their relationship was strengthened by *"going through something pretty bad together"* [45]. While men in both this study and Bilardi's [13] prior study showed high levels of partner support, a number of participants in this study noted they may feel differently if BV were to be classified as an STI.

Different health conditions can be imbued with varied meanings in cultural and social contexts, affecting the way in which they are viewed and experienced [33]. Stigma theory asserts that stigmatisation of illness is not an inherent property of a condition, rather the product of a social process that ascribes meanings to both the condition and by extension, the character of the affected individual as "undesirably different" or "deviant" [46–48]. Stigma is strongly associated with STIs which have been historically constructed as symbols of immoral or irresponsible behaviour [49,50]. Such views have served to perpetuate stereotypes of the "kind of people" who have STIs [28,47,49,51] resulting in a reduction in perceived STI risk if people do not believe they belong to this group [51,52], as well as increased anticipated and internalised stigma if individuals were to contract an STI [47,49,50,53–55]. Such beliefs have been demonstrated to influence sexual health seeking behaviours, psychosocial well-being and self-concept in adjustment to diagnosis, and disruption of intimate relationship dynamics [47,51,54,56,57]. Though not currently classified as an STI, some participants identified that the genital location of BV and the offering of partner treatment could be sufficient to evoke similar constructions of meaning, resulting in treatment avoidance among "other men".

However, several participants themselves demonstrated that the impact of STI stigma can be somewhat attenuated by the current ambiguity of the classification of BV. Men who discussed how they perceived of people who may have STIs and how they may feel with an STI diagnosis did not demonstrate the same response to partner treatment for BV. Furthermore, whilst men did not question the fidelity of their partners' due to their BV diagnosis, this was in part due to an understanding that this condition was *not* an STI. As BV is currently seen as a "lady problem", treatment could be understood as fulfilling the role of the supportive partner, rather than labelling men as having a stigmatising "problem" or introducing questions of fidelity. This finding will be important when considering how clinicians may communicate MPT for BV if it is found to be successful in reducing BV recurrence in women.

What is clear from the results of this study is the central importance of the relationship in influencing not only the impact of BV on men, but their acceptance of treatment. Concern for partners was a unique driving force for treatment acceptance, not merely because men wanted to help ease their partner's suffering, but because they viewed this as their role as a man in a partnership. Most previous BV MPT trials did not collect any information on the nature of the relationships of the participating couples, except for one in which 75% of couples were married

[21]. This overrepresentation of committed couples seems to potentially corroborate the central role of the relationship in determining MPT acceptance for BV.

Participants largely related MPT acceptance to ideas of being a good and supportive partner or a responsible, caring man; whilst they hypothesised that declining treatment may largely be a way for "other" men to maintain a "strong" "masculine" image. In this way, MPT can be understood as tool to demonstrate a certain type of masculine identity.

Participants unanimously identified that it was traditional norms of masculinity among "other" men that would likely represent the *major* barrier for MPT acceptance. Gender relational theory [34] proposes that the responses of men and women to health are based on the gender scripts that determine what is considered to be appropriate behaviour within the specific social structure or culture in which they live [34,58]. Within this theory, images of men as strong and inviolable have been naturalised thereby "feminizing" illness and health care seeking [34,35,58]. Within this gendered script, having what participants termed as "a problem" presents a threat to masculine identity [34,35]. Indeed, theories of gender and health argue that traditional norms of masculinity are one of the most important socio-cultural factors influencing men's health [59,60], attitudes and behaviours in romantic relationships [35] and the associated well-being of their female partners [35,59,60].

However, numerous participants conceived that "being a man" meant being a supportive partner and "taking responsibility". Many simultaneously did this whilst identifying themselves as "different to other men" which demonstrates the ironic observation of Wetherell and Edley (1999) that *"one of the most effective ways of 'being a man' in certain local contexts may be to demonstrate one's distance from a regional hegemonic masculinity"* [61]. This demonstrates the numerous versions of masculinity that can be enacted and revered in different contexts.

A tendency in men's studies is to analyse masculinities by looking only at men and relations among them, presuming that women and their relationships with men are irrelevant [36,62]. However, as all identities are relational, a consistently relational approach should be taken to understanding how individuals "do" gender [36]. Indeed, the validation one receives within their intimate relationships is crucial in defining one's self-concept [43,63], and as such intimate relationships are domains in which men can actively reconstruct their masculine identities [64,65]. Although gender is almost always a background identity that will inform beliefs and behaviours in relational contexts, it is most commonly an individual's other salient identities (such as partner) that shape behaviour and choices in specific situations [66,67].

The central importance of the relationship is also reflected in the STI literature relating to partner notification. While we were unable to identify any studies that explore partner treatment acceptance, the literature demonstrates that men are more likely to notify partners of an STI diagnosis if they are in committed relationships, with stronger emotional ties [29,45,52,68,69]. Indeed, a 2011 US-based study found that people were three-times more likely to inform partners with whom they were in a long-term relationship of an STI diagnosis [70]. This demonstrates that the role of the relationship is an important determinant of sexual health related behaviours and intentions.

Many participants identified that men, including themselves, may refuse MPT for BV due to the belief that treatment would be of no personal benefit. Similarly, a review of literature exploring HPV vaccination acceptability and intention to vaccinate among men demonstrated that this same belief that the vaccine would not directly benefit males was repeatedly cited as the primary reason men would decline HPV vaccination [71] Indeed, one study of college students in the USA demonstrated that only 38% of men would accept vaccination if they understood that this would protect their female partners from cervical cancer alone, compared to 78% that would accept the vaccine if there was the personal benefit of protection against genital

warts [72]. These findings support the finding that the lack of BV symptoms in men may represent a serious barrier to the success of MPT.

However, an unexpected finding of great interest in this study was the prolonged symptoms one participant experienced which he attributed to BV and his resulting belief that BV was sexually transmitted. This participant's feelings of embarrassment, sexual withdrawal and self-stigmatization, and his reported hygiene practices to conceal his symptoms perfectly mirror the documented experiences of women with BV [12–14,44]. His eventual acceptance of his symptoms as somewhat "normal" led him not to seek medical advice, a decision which also reflects female experiences [44]. No prior BV MPT trials have collected data on male partner symptoms [17–24] and literature identifying BVABs on the penis have described participants as asymptomatic and "healthy" [26]. Current knowledge suggests that men do *not* generally experience BV symptoms. However, 30 year old case reports describe three men experiencing *Gardnerella vaginalis*-associated balanoposthitis (malodourous balanitis) [73] and there is very recent interest in whether BVABs may cause urethral symptoms in men [74]. This indicates a growing consideration of how the presence of and exposure to BVABs may physically affect men. Though assessment of urethral symptoms is part of routine sexual health history taking in men, questions of genital malodour are generally only asked of women [75]. This raises questions of how many men may actually be experiencing BV-like symptoms and whether further consideration of this is required in future research.

## 4.1. Strengths and limitations

The major strength of this study is that it is the first study we are aware of to explore male partners' views and experiences of BV and associated MPT, and has identified productive directions for future research.

This study has some very important limitations. Men who had not participated in the StepUp studies may have very different views from the men interviewed in this study. Despite attempting to recruit men who declined MPT through the StepUp studies, none of these men expressed interest in participating. Accordingly, all study participants had accepted MPT. Thus, the reasons identified for MPT decline are largely hypothetical, with the exception of one participant who had previously declined a similar study. As MPT is still experimental and access to it is part of a larger study with its' own requirements, it cannot be said how this may have influenced MPT decline. The lack of cultural diversity among the men interviewed means this study may not have captured the diversity of attitudes or experiences among different populations of men. Male experience of symptoms was an unexpected theme that did not reach saturation and will be the subject of another study.

## 5. Conclusion

Considering the significant health and personal costs of BV on women, and economic costs of repeated infection, evidence of successful sustained cure in women through partner treatment would be a welcome management approach that could radically change the way the infection is managed. However, uptake of treatment by male partners may be heavily influenced by contextual factors that may improve or diminish their willingness to accept treatment. The subjective social norms that underpin how sexual health issues are viewed and how individuals "do gender" will no doubt exert a significant influence over men's decision-making. Encouragingly, it seems that strong caring relationships may mitigate some of these issues, as does the fact BV is not currently classified an STI.

In the absence of male symptoms and a confirmatory test to "prove" the presence of infection in men, achieving MPT acceptance may be more challenging. Interestingly, this research

suggests there may indeed be men suffering from BV like symptoms which may alter their understandings of the personal health benefit of seeking and accepting treatment.

## Supporting information

**S1 Interview schedule.**
(PDF)

## Acknowledgments

We would like to acknowledge the broader StepUp studies research team, and all of the staff at MSHC who facilitated recruitment of couples into the StepUp studies. We would also like to thank the men who kindly participated in this study.

## Author Contributions

**Conceptualization:** Rebecca Wigan, Lenka Vodstrcil, Catriona S. Bradshaw, Jade E. Bilardi.

**Data curation:** Rebecca Wigan, Jade E. Bilardi.

**Formal analysis:** Rebecca Wigan, Cathy Vaughn, Jade E. Bilardi.

**Funding acquisition:** Catriona S. Bradshaw.

**Investigation:** Rebecca Wigan, Jade E. Bilardi.

**Methodology:** Rebecca Wigan, Catriona S. Bradshaw.

**Project administration:** Rebecca Wigan.

**Supervision:** Cathy Vaughn, Jade E. Bilardi.

**Validation:** Jade E. Bilardi.

**Writing – original draft:** Rebecca Wigan, Jade E. Bilardi.

**Writing – review & editing:** Cathy Vaughn, Lenka Vodstrcil, Michelle Doyle, Marti Kaiser, Collette McGuiness, Catriona S. Bradshaw, Jade E. Bilardi.

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
