## [Decision Letter · Decision Letter 0]

14 Apr 2020

PONE-D-20-05638

“It’s just an issue and you deal with it… you just deal with it, you move on and you do it together.”: Male experiences of Bacterial Vaginosis and the acceptability of associated male partner treatment.

PLOS ONE

Dear Dr Wigan,

Thank you for submitting your manuscript to PLOS ONE. After careful consideration, we feel that it has merit but does not fully meet PLOS ONE’s publication criteria as it currently stands. Therefore, we invite you to submit a revised version of the manuscript that addresses the points raised during the review process.

We would appreciate receiving your revised manuscript by 30  May 2020. To enhance the reproducibility of your results, we recommend that if applicable you deposit your laboratory protocols in protocols.io, where a protocol can be assigned its own identifier (DOI) such that it can be cited independently in the future. For instructions see: http://journals.plos.org/plosone/s/submission-guidelines#loc-laboratory-protocols

We look forward to receiving your revised manuscript.

Kind regards,

Tania Crucitti

Academic Editor

PLOS ONE

Journal Requirements:

2. PLOS ONE does not copy edit accepted manuscripts. Therefore, the language in submitted articles must be clear, correct, and unambiguous (http://journals.plos.org/plosone/s/criteria-for-publication#loc-5). We notice that your manuscript contains offensive terms in some of the quotes. We would like you to reduce the number of offensive terms in this manuscript and don't think that these verbatim examples have to be printed to make the points made in your manuscript. Please remove the offensive words on page 16 and 32 and generally make sure that there is only a minimal number of offensive terms in your manuscript and only where absolutely scientifically necessary.

3. Please provide additional details regarding participant consent. In the ethics statement in the Methods and online submission information, please ensure that you have specified how verbal consent was documented and witnessed.

4. Please include additional information regarding the interview guide used in the study and ensure that you have provided sufficient details that others could replicate the analyses. For instance, if you developed a guide as part of this study and it is not under a copyright more restrictive than CC-BY, please include a copy, in both the original language and English, as Supporting Information.

Reviewers' comments:

Reviewer's Responses to Questions

**Comments to the Author**

1. Is the manuscript technically sound, and do the data support the conclusions?

Reviewer #1: Partly

Reviewer #2: Yes

2. Has the statistical analysis been performed appropriately and rigorously? 

Reviewer #1: N/A

Reviewer #2: N/A

3. Have the authors made all data underlying the findings in their manuscript fully available?

Reviewer #1: No

Reviewer #2: No

4. Is the manuscript presented in an intelligible fashion and written in standard English?

Reviewer #1: Yes

Reviewer #2: Yes

5. Review Comments to the Author

Reviewer #1: The author fills an important knowledge gap of male perspectives on their partner’s BV infection, how that affects their relationship with their partner and what attitudes they have about getting treatment for BV, a condition that has typically has no effect on them physically but may give better recovery results for their partners.

Major:

- Ethics section needs more details particularly around confidentiality

- Doubts about maximum variation in participants and data saturation

- Interesting results but need to be presented in a better way

Minor:

- Check for grammar and typos (for example, lines 119, 133, 530…)

Introduction:

The argument is built well and the objective stated clearly.

Minor:

- You mention that BV being sexually transmitted is a controversial and debated idea and then you elaborate more on the possibility of it being sexually transmitted. I would like to see a short sentence summarising what the other lines of thought are if BV is not sexually transmitted, eg ethnic/racial pre-dispositions to BV, BV in women prior to sexual debut ++

Methods:

I liked how reflexivity regarding the interviewer’s gender and profession was considered during the interview process.

Major

- Ethics statements needs more elaboration on confidentiality and verbal consent. How was confidentiality ensured? Was verbal consent taken when setting up an interview time or just before the interview began?

- In Table 1, one of the eligibility criteria listed was “consented to audio recording of interview” Justify this criteria. Why was this important? If it is a sensitive/private topic then perhaps this introduced a bias in the group that agreed to speak with you as perhaps not everyone wants to be audio recorded even though they may not mind participating in the study. If you agree about potentially biased selection of respondents, include in limitations. If you disagree, explain a bit why this was a good approach for the current setting.

- Study setting is missing from methods. All I know is that a hospital in Victoria gave the ethical approval, but where was the study conducted? Later I read about phone interviews, but it needs to be clear in the methods where they were conducted.

- What was the age distribution of the respondents? And ethnicities? How did you ensure maximum variation in the respondents when you purposely selected them? Were your respondents varied enough to conclude saturation?

Minor

- It is unclear to me what you mean by “explore consciousness” on line 113

- Line 177, the acronym JB appears for the first time… who is this person? Why was this person an appropriate person to conclude data saturation? Explain

- You conducted phone interviews about a potentially private/sensitive topic – was anything done to build trust?

- Line 175: Influence on what?

-

Results

You have some interesting and useful findings to fill a knowledge gap.

Major:

- Table 4 is difficult to read, also this isn’t a case study from what I can tell, so presenting the variation as text and explaining it will be better.

- Analysis needs some more work and cleaning up for example, line 231/232: how are 3.1 and 3.2 different? 3.2 seems more about manhood/perceptions of masculinity than “the supportive man” while 4.1.1 also speaks of “manhood” a bit differently.

- Page 16/17: Case 2: “This participant demonstrated extremely high levels of STI stigma and traditional masculinity”. Here ‘stigma’ and ‘traditional masculinity’ need to be contextualised and defined. You do it a bit in your discussion but these concepts can be defined in the methods and theory section.

- Line 650-653: From the data presented here, it seems to me that their sex lives DO matter to them

-

Minor:

- Table 2 “How would they like to receive information” – what information? Specify.

- Line 194, You write about JB and CV being experienced social researchers. This should come earlier when or before JB was first introduced

- Line 200, How were the 20 men chosen to be contacted? Were there only 20 who contacted you with interest in being in a study? When you stopped at 11, did you contact the remaining 9 to inform them why they were not contacted further?

- Table 3. Did you mean “Average length of relationship”?

- The list of themes and sub themes is unnecessary

- Check for repetition in the results

-

Discussion

Major:

- Check for repetition

Minor:

Reviewer #2: This is a really well written article about men's experiences of BV and their views on male partner treatment. It is a qualitative study conducted alongside a RCT where women with confirmed BV were enrolled, as well as male partners.

A few comments:

It would be interesting to know a bit more about what the Step Up Trial consisted of for the male partners, beyond the eligibility requirements. Was it treatment only or were there other aspects of the study (e.g. education).

Consent process: Did the study team require written/electronic informed consent or verbal consent? A bit more detail on the consent process in the text would be helpful.

Recruitment: In the analysis, the authors state that they reached data saturation after 11 participants had been interviewed but, in the results section, it seems that the others who were not interviewed (the remaining 9 of the 20 contacted) were not interested in participating. Had data saturation not been reached, what would have happened?

Methods: 20-60 minute interviews seem quite short for lived experience research, what was the average length?

Results: What was the sexual identification of the 1 participant who did not identify as heterosexual?

Is there a better way to format the case scenarios? It was difficult to read side by side over several pages.

Discussion: Regarding your point about local categories of illnesses and its impact on the disease and treatment experiences (and acceptance), I wonder if there is any literature about the perceptions of men regarding vaccination for HPV and about stigma related to HPV care seeking. This literature, if it exists, might be interesting to explore.

I also think that the finding about BV not being classified as a STI is important to think about when working out how to communicate with the public about treatment and decision making as well as the case of the man who had symptoms.

Additionally, I do think that further research is needed into reasons men decline MPT beyond the hypothetical reasons listed by those who had already accepted it. The group interviewed were a very specific population as they were enrolled in studies about the topic with partners who were also enrolled in studies. Their views might be wildly different than those who refused participation and within the community more broadly but the finding from this study could be used to inform data collection in other studies.

6. PLOS authors have the option to publish the peer review history of their article (what does this mean?). If published, this will include your full peer review and any attached files.

Reviewer #1: No

Reviewer #2: No

---

## [Author Response · Author response to Decision Letter 0]

1 Jun 2020

Response to Reviewers

PONE-D-20-05638

“It’s just an issue and you deal with it… you just deal with it, you move on and you do it together.”: Male experiences of Bacterial Vaginosis and the acceptability of associated male partner treatment.

Thank you to both reviewers for their review of our manuscript. The authors are grateful for their comprehensive and thoughtful review and comments, and for outlining the grammatical inconsistencies in the initial document. Please see below our responses to all raised queries, referenced line numbers refer to the tracked changes document. We trust that you will find all queries satisfactorily addressed and hope you now consider this manuscript suitable for publication. 

Kind regards, 

Rebecca Wigan (on behalf of all authors)

To the editor

1. Comment 

We would like you to reduce the number of offensive terms in this manuscript and don't think that these verbatim examples have to be printed to make the points made in your manuscript. Please remove the offensive words on page 16 and 32 and generally make sure that there is only a minimal number of offensive terms in your manuscript and only where absolutely scientifically necessary.

1. Response

We agree and have removed all words that we believe may have been identified as offensive from the quotes in the manuscript. 

2. Comment

Please provide additional details regarding participant consent. In the ethics statement in the Methods and online submission information, please ensure that you have specified how verbal consent was documented and witnessed.

2. Response

Please see the detailed response to this concern in responses to reviewer one 2a, 2b and 2h. 

3. Comment

Please include additional information regarding the interview guide used in the study and ensure that you have provided sufficient details that others could replicate the analyses. For instance, if you developed a guide as part of this study and it is not under a copyright more restrictive than CC-BY, please include a copy, in both the original language and English, as Supporting Information.

3. Response

We have provided the final iteration of the interview guide as supporting information. 

Reviewer One

1. Introduction

1a) Comment

You mention that BV being sexually transmitted is a controversial and debated idea and then you elaborate more on the possibility of it being sexually transmitted. I would like to see a short sentence summarising what the other lines of thought are if BV is not sexually transmitted, e.g. ethnic/racial pre-dispositions to BV, BV in women prior to sexual debut ++

1a) Response

Thank you for your comment. The following line has been inserted into the introduction (lines 37 to 42) to address this point.

“While there is strong evidence supporting a dominant role for sexual transmission in the pathogenesis of BV, other factors may also contribute, although BV is rare/absent in women prior to sexual debut(1). These factors include racial variants in vaginal microbiota, altered host immunity, an endogenous source, environmental factors such as nutrition and intravaginal practices such as douching (2).”

2. Methods

2a) Comment

Ethics statements needs more elaboration on confidentiality and verbal consent. How was confidentiality ensured? Was verbal consent taken when setting up an interview time or just before the interview began?

2a) Response

Thank you for this comment. We have now elaborated on confidentiality and verbal consent in the Methods, under the recruitment and data collection sections. The following lines have been inserted or revised (in blue).

“RW contacted all men who had indicated their interest, to explain the study in detail and arrange an appropriate time for interview.” (Line 188)

 “Participants were provided with a Participant Information and Consent Form (PICF) prior to the interview which assured them participation was voluntary and that any data they provided would be treated confidentially. Men were informed their data would be stored on double password protected on-site databases, and no identifying information would be used in any publications arising from the study.”

 (Line 194-198)

“All participants chose to participate in a phone interview. Verbal consent was provided whereby RW read the previously provided consent form aloud and asked them to state their name, the date and affirmation of their consent. This was recorded on a separate audio file to their interview data to ensure confidentiality. No names were used during the interview.” (Line 207-211)

2b) Comment

In Table 1, one of the eligibility criteria listed was “consented to audio recording of interview” Justify this criteria. Why was this important? If it is a sensitive/private topic then perhaps this introduced a bias in the group that agreed to speak with you as perhaps not everyone wants to be audio recorded even though they may not mind participating in the study. If you agree about potentially biased selection of respondents, include in limitations. If you disagree, explain a bit why this was a good approach for the current setting.

2b) Response

The reviewer raises a valid point that the requirement for audio recording could potentially bias the group, however, this is our usual process in studies conducted through Melbourne Sexual Health Centre, particularly where sensitive topics exploring lived experiences are concerned. It is important that in interviews such as these, the interviewer is able to pay full attention to participants accounts and pursue relevant lines of questioning without distraction and enable an easy and comfortable flow of conversation. Men were informed when first contacted of the need to audio record the interview for these purposes, and no participants declined upon learning of this requirement. We have provided further justification for this eligibility requirement under “Data collection” with the following lines inserted:

“Men were informed that the interviews needed to be audio recorded to allow the interviewer to pay full attention to their accounts and lived experience of being offered BV treatment. No men raised concerns about the audio recording or declined to participate in the interview based on this requirement. Men were informed they did not need to answer any questions they were not comfortable with and could cease the interview at any point.” (Line 200-205)

2c) Comment

Study setting is missing from methods. All I know is that a hospital in Victoria gave the ethical approval, but where was the study conducted? Later I read about phone interviews, but it needs to be clear in the methods where they were conducted.

2c) Response

All participants were recruited through the Step Up trials which were conducted in and coordinated by the largest free sexual health clinic in Victoria, Australia, the Melbourne Sexual Health Centre (MSHC) (line 193-194). Participants were able to be interviewed in person at MSHC, or via phone (line 202). All participants elected to be interviewed over the phone (line 272). The aforementioned lines were in the submitted manuscript, and we have also added the following line (130) to clarify study location: 

“This study was conducted at Melbourne Sexual Health Centre, Victoria Australia.” (Line 145)

2d) Comment

What was the age distribution of the respondents? And ethnicities? How did you ensure maximum variation in the respondents when you purposely selected them?

2d) Response

Only men with a lived experience of accepting or declining MPT within the StepUp study were eligible to participate as this was central to the research question. As maximum variation sampling is utilised to capture the available diversity of experiences relevant to the research question [1], contacting all eligible men provides the maximum variation possible within this very limited pool of eligible men. Men had to be eligible for participation in the StepUp studies and provide consent to be contacted about a qualitative research study. All men who were eligible to participate were contacted for interview, thereby providing insights from all available angles.

As noted in the paper, participants ranged in age from 23 to 60 (line 273), with a median age of 28 (Table 3). This was not an ethnically diverse sample, as only one participant was born outside of Australia (Table 3) and most identified as Australian with European ancestry. The limited cultural diversity of the sample has been highlighted in the limitations in the previously submitted manuscript.

“The lack of cultural diversity among the men interviewed means this study may not have captured the diversity of attitudes or experiences among different populations of men.” (Line 877-879)

In order to further clarify the limited size of the potential pool from which men could be recruited, and that all eligible men were contacted, the “Recruitment” section of the paper has been changed to “Recruitment and participant selection” (Line 144), the explanation of who was contacted and recruited has been altered as below.

“During the recruitment period, before it was decided data saturation had been met, 20 men were eligible for this study. All eligible men were contacted for interview. Of those contacted, 11 men were interviewed; 7 were recruited from the StepUpPilot and 4 from the StepUpRCT. Of the remaining 9 men that were contacted, 8 did not respond to the SMS invitation and 1 decided to withdraw after consenting. These 9 men were not followed up further as per the study protocol and ethics approval. If data saturation had not been met after 11 interviews, further participants would have been prospectively recruited from the men enrolled in the ongoing StepUp RCT.” (Line 255-263)

The following lines have also been amended (in blue) and added to the manuscript to improve clarity and address the questions about participants:

 “All men who were still actively engaged in the StepUp pilot at the time of ethics approval were contacted via a single SMS offering the opportunity to participate in the qualitative study.” (Line 171-172)

“All men who provided consent were contacted by RW until saturation was deemed to be met.” (Line 179-180)

2e) Comment

Were your respondents varied enough to conclude saturation?

2e) Response

In addition to our response to Comment 2d above, saturation was reached within the available sample we were able to recruit. As noted in our earlier response, maximum diversity was limited but achieved among the eligible men who were offered MPT for BV and expressed interest in the qualitative study. Male experience of symptoms was an unexpected theme that did not reach saturation, and will be the subject of another study. We have also added the inability to reach saturation on this theme in the limitations.

“Male experience of symptoms was an unexpected theme that did not reach saturation, and will be the subject of another study” (Line 879-881)

2f) Comment

It is unclear to me what you mean by “explore consciousness” on line 113

2f) Response

 We have replaced “explore consciousness” to “explore lived experiences” in line 128 to improve clarity for the reader. 

2g) Comment

Line 177, the acronym JB appears for the first time… who is this person? Why was this person an appropriate person to conclude data saturation? Explain

2g) Response

Thank you for raising this. We have amended this in the manuscript by introducing relevant members of the research team (JB and CV) at an earlier point in the paper. 

“RW met regularly with JB and CV, experienced social researchers adept in qualitative methods in the area of sexual and reproductive health, to discuss the interviews and emerging themes.” (Line 226-227)

2h) Comment

You conducted phone interviews about a potentially private/sensitive topic – was anything done to build trust?

2h) Response

RW is an experienced sexual health nurse accustomed to discussing sexual health matters with patients and participants and made a considerable effort to put participants at ease. Participants were also given multiple opportunities to ask questions, to decline participation or cease the interview. Men were reminded in the PICF and at the beginning of the interview that they did not need to answer anything they did not feel comfortable with. As participants were either actively participating in Step Up RCT, or had recently participated, they had developed trust in the research teams at Melbourne Sexual Health Centre. 

We have added the following line to the manuscript to clarify this in the text also:

“Men were informed they did not need to answer any questions they were not comfortable with and could cease the interview at any point.” (Line 204-205)

2i) Comment

Line 175: Influence on what?

2i) Response

Thank you for highlighting this. We have clarified this in line 230.

“…including questions around the influence of uncertainty of infection in men on their willingness to accept MPT.”

3. Results

3a) Comment

Table 4 is difficult to read, also this isn’t a case study from what I can tell, so presenting the variation as text and explaining it will be better.

3a) Response

This table is intended to provide a snapshot of the complexity and differences in experiences among men that we feel is better achieved in a table than in text. We have provided a similar 3 case scenario table in a previous PLOS One paper exploring women’s experiences of BV and it was well received in the paper [2]. The case study format allows the provision of a summary of a single individuals overall experience and relates to a number of themes, particularly in case study 3 (who declined study participation in a similar trial years prior with the same partner, and accepted on this occasion). It is the authors preference to keep these descriptions in table format if possible but we have reformatted the table for easier readability. We have also reduced the wordcount within the case studies to keep the table as small as possible without losing the narrative of these participants experiences, which is intended to provide the reader with a clearer understanding of the diversity of men’s lived experiences across themes presented in the manuscript. 

If the reviewers/editor feels that the table of case scenarios do not add any value to the paper, we can remove this table from the manuscript. Please note, if it is removed, we would need to reformat some aspects of the manuscript where the reader is referred to the cases studies.

The reformatted table (Line 312) currently occupies only one page if it is not embedded within text. 

3b) Comment

Analysis needs some more work and cleaning up for example, line 231/232: how are 3.1 and 3.2 different? 3.2 seems more about manhood/perceptions of masculinity than “the supportive man” while 4.1.1 also speaks of “manhood” a bit differently.

3b) Response

Thank you for this feedback. We can see how 3.1 and 3.2 may be difficult to differentiate. 

3.1 Speaks to the role of the relationship itself in influencing acceptance, whilst 3.2 speaks to men’s perceptions of what it is to be a man in relation to their partner or women more generally. 4.1.1 speaks of manhood in a different way as it is presenting the idea of the “masculine” version of manhood as a potential barrier to acceptance. We pick up on these differences throughout the discussion. 

To improve the clarity of distinction between 3.1 and 3.2, we have changed the subtitle of 3.2 from “The supportive man” to “My responsibility as a man” (Line 531). We have also removed a portion of 3.1 that could be viewed as repetitive (Lines 508-515). 

We feel that this feedback has significantly improved the clarity of these themes. Thank you. 

3c) Comment

Page 16/17: Case 2: “This participant demonstrated extremely high levels of STI stigma and traditional masculinity”. Here ‘stigma’ and ‘traditional masculinity’ need to be contextualised and defined. You do it a bit in your discussion but these concepts can be defined in the methods and theory section.

3c) Response

Thank you for raising this point. In order to clarify the meaning of this in context, we have changed the wording in this case study as highlighted in blue below:

“Despite accepting treatment, this participant demonstrated an extremely judgmental attitude towards STIs and those who had experienced them, as well as demonstrating high levels of the type of “manly” masculinity described by all participants as a likely barrier to MPT acceptance. “(Table Four)

We have also added the lines below to the theory section of the manuscript to highlight the role of norms of masculinity earlier in the manuscript.

“Theories of gender and health argue that health-related beliefs and behaviours are a means for demonstrating masculinity and are therefore culturally understood symbols that can be leveraged to assert a particular identity or cultural norm (5–7) Traditional masculine ideals that prescribe stoicism, dominance, independence and self‐reliance (5–8) are increasingly identified as one of the most important socio-cultural factors influencing men’s health attitudes, behaviours and outcomes.” (Line 119-124)

3d) Comment

Line 650-653: From the data presented here, it seems to me that their sex lives DO matter to them

3d) Response

Thank you for your comment. The presence of female symptoms does not bother men as much as their partners response to them (embarrassment, shame and reduced self-esteem). Accordingly, we have slightly changed the wording of this line to improve clarity. The change is underlined below. 

“This study has demonstrated that while the symptoms of BV often do have an impact on men’s sex lives this appears to be mainly due to the shame and embarrassment women feel, and men in this study demonstrated they are more concerned with the impact of symptoms on their partners self-esteem than their sex lives per se.” (Line 719-723)

3e) Comment

Table 2 “How would they like to receive information” – what information? Specify.

3e) Response

This line has been changed to the following (change in blue):

“How would they like to receive information about BV” (Table 2)

3f) Comment

Line 194, You write about JB and CV being experienced social researchers. This should come earlier when or before JB was first introduced

3f) Response

Thank you for highlighting this. We have amended this in the manuscript by introducing JB and CV earlier in the paper (please see earlier response 2g).

3g) Comment

Line 200, How were the 20 men chosen to be contacted? Were there only 20 who contacted you with interest in being in a study? When you stopped at 11, did you contact the remaining 9 to inform them why they were not contacted further?

3g) Response

As described in the response addressing maximum variation (2d), we have amended the manuscript to clarify that 20 men were eligible and all were contacted by RW.

Those who did not reply to the initial invitation SMS or withdrew consent were not followed up further as per study protocol and HREC approval. We have also amended the results section to clarify these remaining men did not respond to the invitation to participate and were therefore not followed up further. 

“During the recruitment period, before it was decided data saturation had been met, 20 men were eligible for this study. All eligible men were contacted for interview. Of those contacted, 11 men were interviewed; 7 were recruited from the StepUp Pilot and 4 from the StepUp RCT. Of the remaining 9 men that were contacted, 8 did not respond to the SMS invitation and 1 decided to withdraw after consenting. These 9 men were not followed up further as per the study protocol and ethics approval.” (Line 260-261)

3f) Comment

Table 3. Did you mean “Average length of relationship”?

3f) Response

This is the median length of relationship, and the range. We agree it is currently made difficult for the reader by the table being split across two pages due to insertion directly under the paragraph in which it was mentioned as per submission guidelines. 

3g) Comment

The list of themes and sub themes is unnecessary

3g) Response

Thank you for this feedback. We have removed this list. 

3h) Comment

Check for repetition in the results

3h) Response

We believe that this may refer to the perceived repetition in the results in 3.1 and 3.2 and have addressed this in our earlier comment (3b). We have now clarified the difference between these two themes and removed some of the text that may have been viewed as repetitive.

Reviewer Two

1a) Comment

It would be interesting to know a bit more about what the Step Up Trial consisted of for the male partners, beyond the eligibility requirements. Was it treatment only or were there other aspects of the study (e.g. education).

1a) Response

Thank you for this feedback. We have added the below to describe the requirements of the Step Up Pilot and RCT within the manuscript. 

“Within the Step Up studies, male partners of women being treated for BV were randomised to oral Metronidazole and a topical antibiotic cream applied to the penis twice a day for 7 days or current standard of care (female treatment only). Participants self-collected genital samples monthly for a follow up period of 3 months. All male procedures, including initial recruitment were able to be conducted remotely via phone and using post packs. To ensure consistency, all male partners were provided with the same information regarding the trial and BV at recruitment; that BV is a commonly occurring infection in women, that BV associated bacteria have been found on male genitals and that these bacteria may be exchanged during sex. No further education was provided. Men were also made aware that all answers in questionnaires were kept strictly confidential and not shared with partners.” (Line 150-160)

2a) Comment

Consent process: Did the study team require written/electronic informed consent or verbal consent? A bit more detail on the consent process in the text would be helpful.

2a) Response

Thank you for this feedback. We have addressed this point earlier in the response to Reviewer 1’s comments under subheading 2a.

3a) Comment 

In the analysis, the authors state that they reached data saturation after 11 participants had been interviewed but, in the results section, it seems that the others who were not interviewed (the remaining 9 of the 20 contacted) were not interested in participating. Had data saturation not been reached, what would have happened?

3a) Response 

Thank you for this question. Had data saturation not been met within this sample, interviews would have continued as participants were recruited to the ongoing Step Up RCT. We have added the line below to the manuscript to clarify this for the reader. 

“If data saturation had not been met after 11 interviews, further participants would have been prospectively recruited from the men enrolled in the ongoing StepUp RCT.” (Line 262-264)

One area where saturation was not met was the experience of men with symptoms they attributed to BV. This was an unexpected theme and beyond the scope of the original research question and therefor will be the subject of a future paper. We have just received ethics approval to conduct interviews to explore symptomatic men’s experiences further.

For further elaboration on the limited ability to purposively sample for maximum variation, we invite the reviewer to see our response to Reviewer 1’s comment under subheading 2d. 

4. Methods

4a) Comment

20-60 minute interviews seem quite short for lived experience research, what was the average length?

4a) Response

The average length was 39 minutes. We have added this to the manuscript:

“All participants chose the option of a telephone interview, which lasted between 20 to 60 minutes (average 39 minutes).” (Line 274)

4b) Comment

What was the sexual identification of the 1 participant who did not identify as heterosexual?

4b) response

This participant identified as bicurious, which is a heterosexual person who is interested in having a sexual experience with a person of the same sex. 

This information and definition has been added to Table 3.

4c) Comment

Is there a better way to format the case scenarios? It was difficult to read side by side over several pages.

4c) Response

Thank you for this feedback. As per our earlier response to Reviewer 1, we have reformatted the table to improve readability.

4d) Comment

Regarding your point about local categories of illnesses and its impact on the disease and treatment experiences (and acceptance), I wonder if there is any literature about the perceptions of men regarding vaccination for HPV and about stigma related to HPV care seeking. This literature, if it exists, might be interesting to explore.

4d) Response

The reviewer’s comment inspired us to look at the literature regarding willingness of men to accept HPV vaccination in comparison to women, as this is recognised (among heterosexuals in particular) to be of greatest benefit for females. Therefore, there appear to be parallels between men’s willingness to accept MPT “for his partner” and accept HPV vaccination “to protect women”. We would like to thank the reviewer for this comment at this is an interesting parallel we had not identified, and the literature appears to reflect our own finding that the lack of perceived personal benefit is of great importance. We have therefore added the following lines to the discussion of our paper. 

“Many participants identified that men, including themselves, may refuse MPT for BV due to the belief that treatment would be of no personal benefit. Similarly, a review of literature exploring HPV vaccination acceptability and intention to vaccinate among men demonstrated that this same belief that the vaccine would not directly benefit males was repeatedly cited as the primary reason men would decline HPV vaccination (9) Indeed, one study of college students in the USA demonstrated that only 38% of men would accept vaccination if they understood that this would protect their female partners from cervical cancer alone, compared to 78% that would accept the vaccine if there was the personal benefit of protection against genital warts (10). These findings support the finding that the lack of BV symptoms in men may represent a serious barrier to the success of MPT.” (Line 834-844)

4e) Comment

I also think that the finding about BV not being classified as a STI is important to think about when working out how to communicate with the public about treatment and decision making as well as the case of the man who had symptoms.

 4e) Response

We also feel that this is a very important point for future communications regarding BV treatment. We have also added the line below to more clearly identify the importance of this finding.

“This finding will be important when considering how clinicians may communicate MPT for BV if it is found to be successful in reducing BV recurrence in women.” (Line 773-734)

As noted in response to reviewer one (2e) we plan to undertake further interviews specifically with men who feel they have had BV-like symptoms in a separate study.

References in this response

1. Etikan I, Musa SA, Alkassim RS. Comparison of Convenience Sampling and Purposive Sampling. Am J Theor Appl Stat. 2015 Dec 22;5(1):1. 

2 Bilardi JE, Walker S, Temple-Smith M, McNair R, Mooney-Somers J, Bellhouse C, et al. The Burden of Bacterial Vaginosis: Women’s Experience of the Physical, Emotional, Sexual and Social Impact of Living with Recurrent Bacterial Vaginosis. Ratner AJ, editor. PLoS ONE. 2013 Sep 11;8(9):e74378.

---

## [Editor Report · Decision Letter 1]

12 Jun 2020

“It’s just an issue and you deal with it… you just deal with it, you move on and you do it together.”: Male experiences of Bacterial Vaginosis and the acceptability of associated male partner treatment.

PONE-D-20-05638R1

Dear Dr. Wigan,

We’re pleased to inform you that your manuscript has been judged scientifically suitable for publication and will be formally accepted for publication once it meets all outstanding technical requirements.

Kind regards,

Tania Crucitti

Academic Editor

PLOS ONE

Additional Editor Comments (optional):

A typo occured in line 55 Bactria should read Bacteria, line 723 partners should read partners'

Please also check that names of bacteria are written in italic, for example line 857 Gardnerella vaginalis
---

## [Editor Report · Acceptance letter]

17 Jun 2020

PONE-D-20-05638R1 

“*It’s just an issue and you deal with it… you just deal with it, you move on and you do it together*.”: Men’s experiences of Bacterial Vaginosis and the acceptability of male partner treatment 

Dear Dr. Wigan:

I'm pleased to inform you that your manuscript has been deemed suitable for publication in PLOS ONE. Congratulations! Your manuscript is now with our production department. 

Kind regards, 

on behalf of

Dr. Tania Crucitti 

Academic Editor

PLOS ONE